# Collateral activity of the CRISPR/RfxCas13d system in human cells

Peiguo Shi[1✉], Michael R. Murphy[1], Alexis O. Aparicio [1], Jordan S. Kesner [1], Zhou Fang[1], Ziheng Chen [1,2], Aditi Trehan[1], Yang Guo[1] & Xuebing Wu [1✉]

CRISPR/Cas13 systems are increasingly used for programmable targeting of RNAs. While Cas13 nucleases are capable of degrading both target RNAs and bystander RNAs in vitro and in bacteria, initial studies fail to detect collateral degradation of non-target RNAs in eukaryotic cells. Here we show that RfxCas13d, also known as CasRx, a widely used Cas13 system, can cause collateral transcriptome destruction when targeting abundant reporter RNA and endogenous RNAs, resulting in proliferation defect in target cells. While these results call for caution of using RfxCas13d for targeted RNA knockdown, we demonstrated that the collateral activity can be harnessed for selective depletion of a specific cell population defined by a marker RNA in an in vitro setting.

[1] Department of Medicine and Department of Systems Biology, Columbia University Irving Medical Center, New York, NY, USA. [2] Present address: Department of Biological Sciences, Carnegie Mellon University, Pittsburgh, PA, USA. ✉email: ps3008@cumc.columbia.edu; xw2629@cumc.columbia.edu

CRISPR/Cas are prokaryotic adaptive immune systems against viruses[1]. In these systems, Cas effector proteins, typically nucleases, are loaded with programmable small guide RNAs (gRNAs) and recruited to targets via sequence complementarity. DNA-targeting CRISPR systems, including Cas9 and Cas12, have been repurposed for engineering of the genome and the epigenome in both basic research and therapeutic applications[2]. Similarly, multiple RNA targeting CRISPR systems have also been identified, including Cas13 that have been widely used for targeted binding or cleavage of RNAs in eukaryotic cells[3–6].

A unique feature of Cas13 systems is the collateral activity. In bacteria, upon sequence-specific recognition of viral RNAs, the non-specific RNase activity of Cas13 becomes activated and degrades both target and non-target RNAs indiscriminately. Mechanistically, target RNA binding induces Cas13 conformation change, which activates an endoribonuclease activity on the surface of Cas13, resulting in cleavage of both the target RNA and bystander RNAs[4]. While such collateral activity has been confirmed for all Cas13 proteins tested in vitro[3–6], and has been harnessed for sensitive detection of nucleic acids[7], early investigations found no such collateral activity in eukaryotic cells[3,6,8]. Furthermore, subsequent studies reported conflicting results[9–12], in part due to the difficulty in separating collateral activity from off-target activity caused by partial matches to the guide RNA (gRNA), as well as the lack of a proper internal control when, in principle, every cellular RNA is downregulated by the collateral activity.

During testing of the RfxCas13d system, a widely used Cas13 effector also known as CasRx[6], we unexpectedly observed strong collateral activities in human cells. Further characterization including RNA-seq with spike-in controls confirmed that targeting abundant RNAs, both exogenously expressed reporters and endogenous RNAs, caused strong collateral degradation of other RNAs in cells, resulting in cell proliferation defect. In a proof-of-principle experiment, we show that RfxCas13d collateral activity can be leveraged to deplete a specific cell type by targeting a dispensable marker RNA.

## Results

**Targeting abundant reporter RNAs causes collateral degradation of the transcriptome in human cells.** We tested RfxCas13d for knocking down a DsRed reporter in human HEK293T cells. When RfxCas13d, DsRed, and a DsRed-targeting gRNA were co-transfected into HEK293T cells, we unexpectedly found that RfxCas13d itself was also markedly downregulated, as indicated by the loss of GFP that is encoded by the same mRNA as RfxCas13d (RfxCas13d-2A-GFP) (Fig. 1a, b). The loss of Cas13d was confirmed by both qPCR (Fig. 1c) and Western blotting (Fig. 1d). Another non-target, co-transfected BFP, was also downregulated by 80% (Fig. 1a–c). The loss of either BFP or GFP is entirely dependent on the presence of the targeted RNA (i.e., DsRed) (Supplementary Fig. 1), ruling out the possibility that BFP and GFP are off-targets with partial match to the DsRed gRNA. Considerable variation in collateral GFP degradation was observed across ten gRNAs targeting different sites in DsRed reporter mRNA, a variation largely explained by on-target DsRed knockdown efficiency (Pearson correlation coefficient r = 0.89, Fig. 1e). Similarly, reducing the expression of either RfxCas13d or the gRNA weakens both the collateral activity and the on-target knockdown efficiency (Supplementary Fig. 2a–e), consistent with these two activities being carried out by the same active site on the Cas13 surface[4]. Taken together, these results demonstrated collateral degradation of bystander RNAs by the RfxCas13d system in human cells upon recognition of a target RNA.

Globally, the collateral activity induced by DsRed-targeting led to a 46% reduction in total RNA (Fig. 1f), which is predominantly composed of ribosomal RNAs (rRNAs). Accordingly, we observed rRNA fragmentation and a significant reduction of RNA integrity (Supplementary Fig. 3). To systematically quantify the impact of RfxCas13d collateral activity on the transcriptome, we performed poly(A) RNA-seq on HEK293T cells using the DsRed-targeting RfxCas13d system. To account for any global reduction in RNA abundance, we used an equal number of cells in each sample and spiked in 5% untransfected mouse MEF-1 cells prior to RNA extraction and sequencing. When normalized to the spike-in mouse RNAs, almost the entire human transcriptome was downregulated by half (median decrease of 46%), including the commonly used internal controls GAPDH and ACTB (Fig. 1g). This near-uniform decrease in rRNA and mRNA abundance confirmed the lack of specificity for the collateral activity of RfxCas13d and underscored the difficulty in detecting collateral activity without a spike-in control. Consistent with the strong knockdown at the protein level (Fig. 1a–d), we observed a very strong decrease of the target DsRed RNA (95%) (Fig. 1g). Notably, the co-transfected non-target BFP and RfxCas13d/GFP RNAs were also downregulated more strongly than endogenous RNAs (90% and 85%, respectively, compared to 46%). Interestingly, human mitochondrial RNAs as a group are less affected by the collateral activity (median 14% knockdown, cyan in Fig. 1g), presumably because these RNAs are shielded by the mitochondrial membrane and thus may be used as internal controls. Taken together, our spike-in based RNA-seq quantification revealed the transcriptome-wide impact of RfxCas13d collateral activities in human cells.

Our observation of strong collateral RNA degradation by RfxCas13d is unexpected, given that the same RfxCas13d system was previously shown to be extremely specific in mammalian cells, with no off-target activity detected by RNA-seq when targeting two endogenous genes (ANXA4 and B4GALNT1)[6]. However, we noticed that both targets are lowly expressed (<1% of GAPDH). When DsRed was expressed at a level similar to ANXA4 and B4GALNT1, no systematic down-regulation of the transcriptome was observed by RNA-seq analysis (Fig. 1h). As each target RNA molecule can only activate a single RfxCas13d/gRNA complex, collateral damage to the transcriptome is expected to scale with the abundance of the target. Indeed, using a serial dilution of the target (DsRed plasmid), we found that the collateral activity (as indicated by BFP loss) is strongly correlated with target abundance (Spearman correlation coefficient r = 0.99, Fig. 1i, j, and Supplementary Fig. 2f).

**Targeting abundant endogenous transcripts also causes strong collateral effects.** We next targeted 11 endogenous mRNAs expressed at various levels, focusing on genes that are non-essential in most RNAi screens (DepMap[13]) (Fig. 2a and Supplementary Fig. 4). Targeting four highly expressed genes whose expressions are comparable to GAPDH led to a marked decrease of RfxCas13d/GFP (i.e., strong collateral activity) (Fig. 2b, c). Similar to DsRed-targeting, collateral degradation of GFP is positively correlated with on-target knockdown efficiency (Supplementary Fig. 4e–m). In contrast, targeting three lowly expressed genes that are at ~1% of GAPDH, including ANXA4 and B4GALNT1 tested in the original RfxCas13d study[6], showed no apparent decrease of GFP (Fig. 2b, c). Targeting four medium abundance genes with expression about 10% of GAPDH resulted in moderate loss of GFP (Fig. 2b, c). RNA-seq analysis further confirmed target abundance-dependent transcriptome-wide down-regulation (Fig. 2d), with 24%, 15%, and 8% decrease of the transcriptome (relative to mitochondrial RNAs) when targeting a high-, medium-, and low-abundance mRNA (HNRNPA2B1, CD99, and B4GALNT1), respectively. The transcriptome-wide destruction triggered by the recognition of a single target

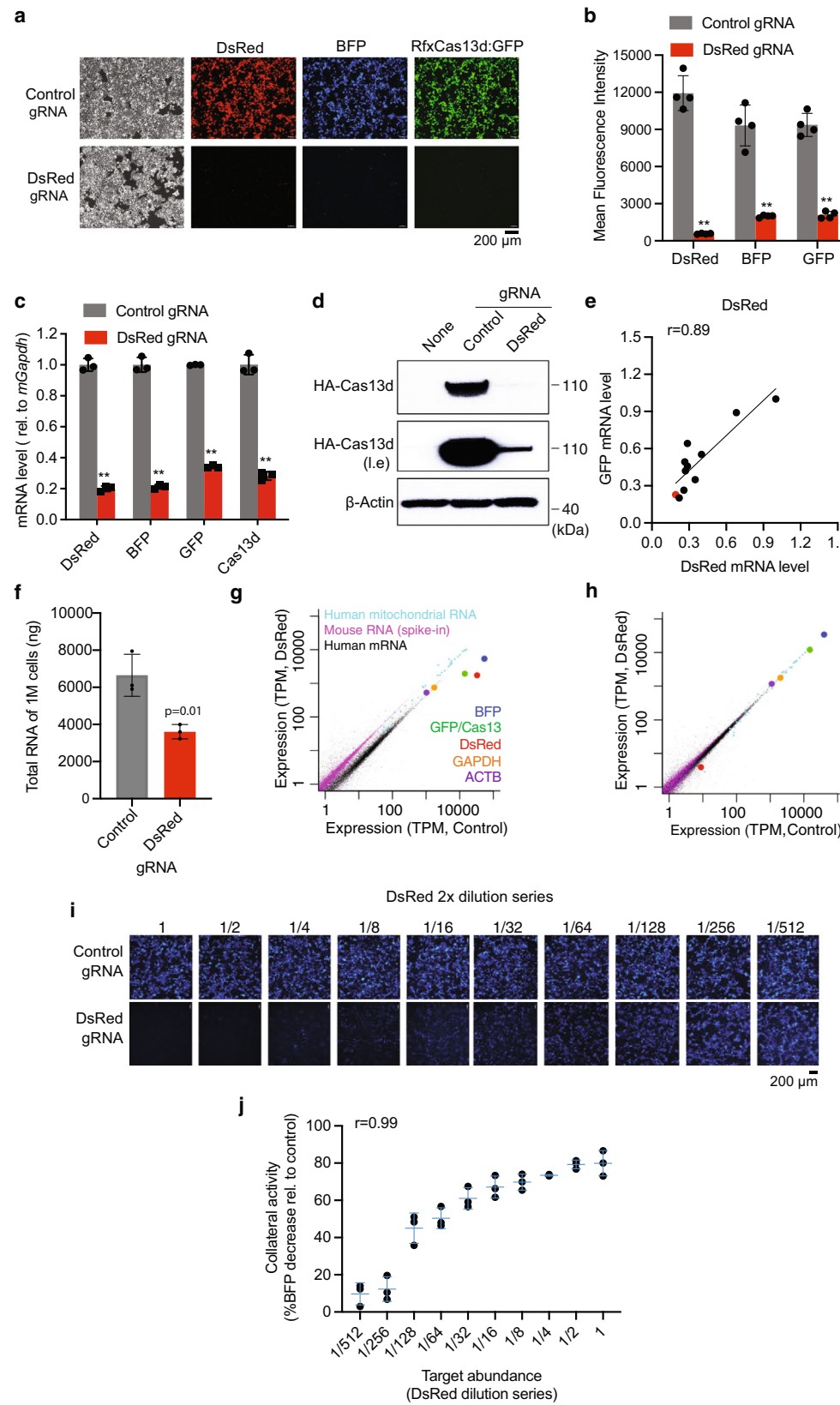

underscored the fact that Cas13 is a potent multiple-turnover enzyme, with each active molecule capable of cleaving 10,000 substrates[5]. As negative controls, similar RNA-seq analyses of siRNA-based knockdown of *HNRNPA2B1* or another abundant mRNA *HNRNPL* did not result in significant transcriptome down-regulation (Supplementary Fig. 5).

**RfxCas13d collateral activity inhibits cell proliferation**. In bacteria, CRISPR/Cas13-mediated destruction of the transcriptome is known to result in host dormancy when induced by viral infection[14]. Similarly, we observed that collateral activity induced by targeting of DsRed strongly affects cell proliferation (as measured by WST-1 assay, Fig. 3a) in a target (i.e., DsRed)

**Fig. 1 Targeting a DsRed reporter mRNA with CRISPR/RfxCas13d causes collateral activity in human cells.** HEK293T cells were transfected with DsRed, BFP, RfxCas13d-2A-GFP, and either a control gRNA or a DsRed-targeting gRNA. (**a**) representative fluorescence imaging data. (**b**) quantification of (**a**) ($N = 4$). (**c**) qRT-PCR ($N = 3$). (**d**) Western blot with anti-HA. (**e**), scatter plot of DsRed and GFP mRNA levels measured by qRT-PCR for ten gRNAs targeting DsRed. Red dot indicates the gRNA used throughout this study. (**f**) total RNA quantified by NanoDrop ($N = 3$). **g, h** poly(A) RNA-seq (**g** $N = 3$, **h** $N = 2$, replicates were pooled). For qPCR and RNA-seq, 5% mouse MEF-1 cells were spiked into equal number of HEK293T cells prior to RNA extraction. Mouse Gapdh ($mGapdh$) was used as load control in (**c**). 5,000-fold less DsRed plasmid was used in (**h**). (**i**) BFP signal when a 2-fold dilution series of DsRed was used (quantified in **j** $N = 3$). Two-tailed Student's $t$-test was used for (**b, c, f**). $**p < 0.01$. l.e. long exposure. Error bars represent standard deviation.

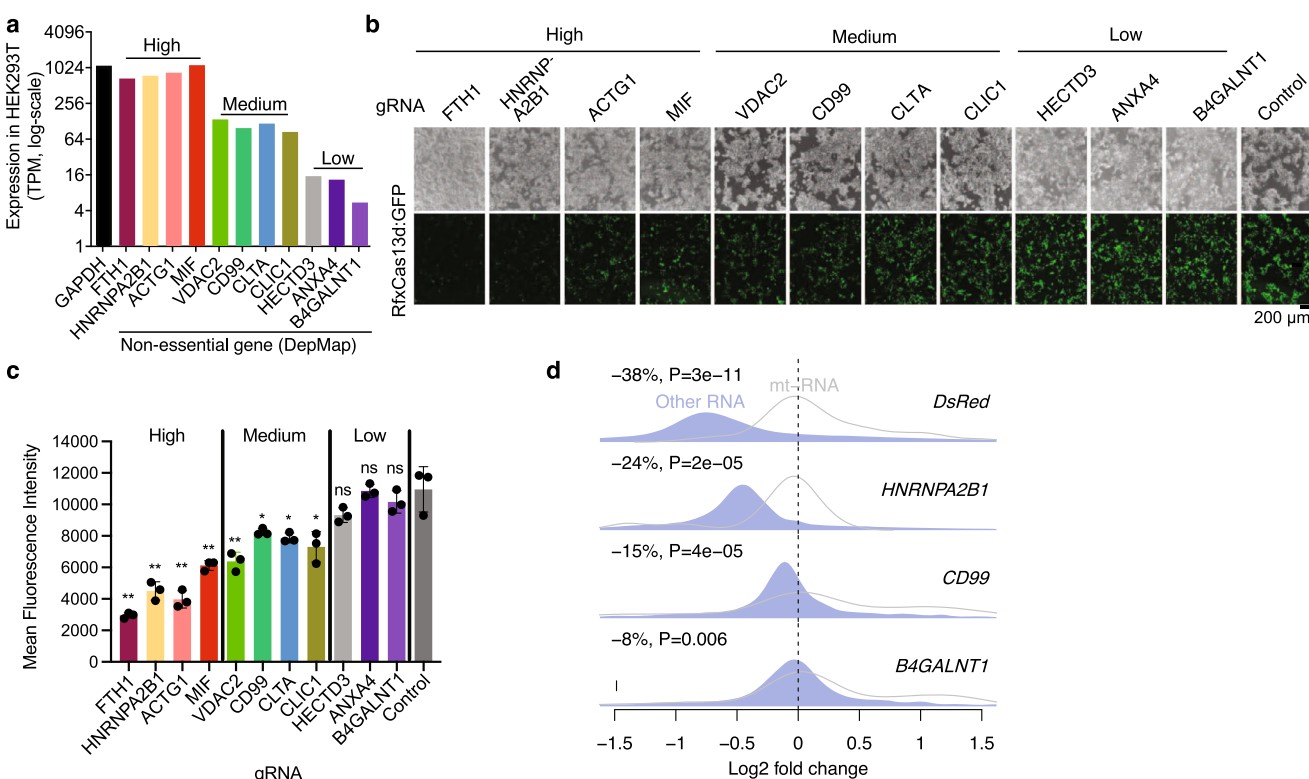

**Fig. 2 CRISPR/RfxCas13d collateral activity triggered by endogenous RNA targeting. a** Expression levels of tested endogenous genes in HEK293T cells. **b** Representative image of GFP when each gene was targeted individually. **c** Quantification of GFP in (**b**, $N = 3$). **d** Density plots of log2 fold changes (knockdown vs. control) of all transcripts normalized to mitochondrial RNAs (RNA-seq). Median down-regulation (%) and the $P$ value are shown at the top left corner of each panel. Two replicates were pooled for (**d**). Two-way ANOVA was used for (**c**). Wilcoxon Rank Sum Test was used for (**d**). $*p < 0.05$; $**p < 0.01$. TPM tags per million. Error bars represent standard deviation.

abundance-dependent manner (Supplementary Fig. 6) without causing significant cell death, apoptosis, or autophagy (Supplementary Fig. 7). Compared to controls, targeted cells showed a marked decrease in DNA replication activity (reduced EdU incorporation, Fig. 3b). Intriguingly, unlike the nuclei in control cells that have a smooth outline and more uniformly distributed chromatin, the nuclei of DsRed-targeted cells often have a rugged and irregular boundary and contain dense chromatin clumps (Fig. 3c, d). This chromatin collapse phenotype is similar to that of cells treated with RNase[15,16], consistent with activated RfxCas13d functioning as a non-specific RNase. In addition to impairing DNA replication, the collapse of chromatin may also inhibit global transcription, which potentially underlies the heightened down-regulation of genes that only become transcribed in the presence of collateral activity (e.g., *GFP* and *BFP*). Targeting the four high-abundance mRNAs mentioned previously led to a similar reduction in cell proliferation (Fig. 3e). In contrast, siRNA-mediated knockdown of the same mRNAs did not affect cell proliferation despite higher knockdown efficiencies than RfxCas13d (Supplementary Fig. 8), supporting that RfxCas13d collateral activity, rather than on-target knockdown, is

responsible for the proliferation defect. As expected, targeting mRNAs of medium abundance resulted in a moderate decrease in cell growth and no significant change for low-abundance targets (Fig. 3e).

The proliferation defect in Cas13 targeted cells suggests that collateral activity can potentially be used to deplete a specific cell population defined by a marker RNA that may have no functional role in target cells. While previous studies have shown that targeting oncogenic mRNAs such as $KRAS^{G12D}$ in pancreatic cancer[17] and $EGFRvIII$ in glioblastoma[9] cause growth defect in cancer cells, it remains unclear whether the defect was due to the loss of the addicted oncogenic mRNA or due to Cas13-collateral activity. To demonstrate the utility of RfxCas13d in selectively depleting a sub-population of cells expressing a specific marker RNA that has no functional role in the cells, we generated a human U87 glioblastoma cell line stably expressing RfxCas13d (no GFP). We subsequently derived two U87(RfxCas13d) cell lines stably express either GFP (U87$^{GFP}$) or BFP (U87$^{BFP}$). The GFP-expressing cells additionally express the target marker RNA DsRed. We then mixed the two cell lines and expressed either the control gRNA or the DsRed-targeting gRNA in the cell mixture

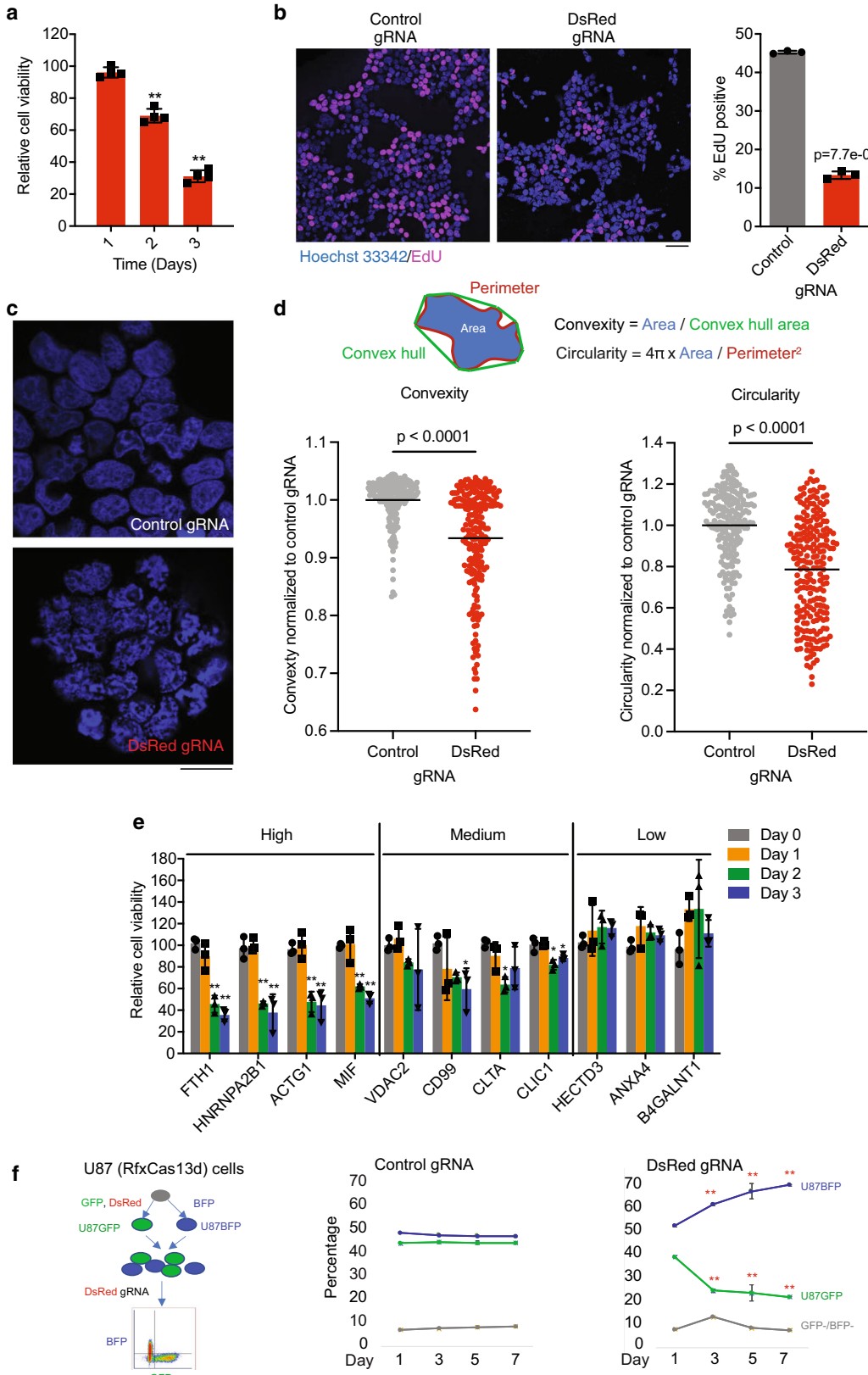

(Fig. 3f). In this competitive growth assay, the fraction of the targeted U87$^{GFP}$ cells decreased over time, whereas U87$^{BFP}$ cells increased (Fig. 3f). These results demonstrated that the collateral activity of RfxCas13d can be harnessed for sequence-specific inhibition of cell proliferation, regardless of whether or not the target RNA plays any functional role in the cell.

## Discussion

In this study, we showed that the CRISPR/RfxCas13d system can exhibit collateral activity in human cells in a target abundance-dependent manner, resulting in proliferation defect in targeted cells. The collateral activity is stronger with higher expression of target RNAs, Cas13 protein, and guide RNA, especially highly

**Fig. 3 Harnessing CRISPR/RfxCas13d collateral activity for RNA-guided cell targeting. a–d** HEK293T cells were transfected with RfxCas13d, DsRed, and either a DsRed-targeting gRNA or a non-targeting control gRNA. Shown are (**a**) cell proliferation (relative to control gRNA) as measured by WST-1 assay, (**b**) EdU incorporation assay, (**c**) nuclear morphology with DAPI staining, and (**d**) quantification of (**c**). **e** WST-1 assay for targeting endogenous mRNAs. (**f**), Selective depletion of DsRed-expressing U87$^{GFP}$ cells from a mixture of U87$^{GFP}$ and U87$^{BFP}$ cells by using RfxCas13d (stably expressed) and a control gRNA or DsRed gRNA (transfected on day 0). Two-way ANOVA was used for (**a**, **e**, **f**) and two-tailed Student's $t$-test was used for (**b**, **d**). $*p < 0.05$; $**p < 0.01$. $N = 3$–4 for (**a**, **b**, **e**, **f**). 208 cells from the control group and 224 cells from DsRed-targeting gRNA group were measured for (**d**). Scale bar, 50 μm (**b**, **c**). Error bars represent standard deviation.

efficient guides. While these results call for caution in using CRISPR/RfxCas13d for targeted RNA knockdown, we demonstrated that such collateral activity can be harnessed to deplete a specific cell population defined by a marker RNA. Expressed maker RNAs are commonly used to define cell types and states, including pathogenic cell states like cancer cells. However, it remains challenging to target and perturb specific cell states based on these marker RNAs that are often dispensable for maintaining the cell state. Further development of a CRISPR/Cas13-based programmable RNA-guided cell targeting platform has the potential to transform functional studies of the myriad cell types and cell states uncovered by deep transcriptomic surveys. It may also advance therapeutic development by allowing sequence-specific targeting of cancer cell proliferation, immune cell inflammatory state, and other pathogenic cell types and states.

During peer-review of this manuscript, the collateral activity of RfxCas13d in mammalian cells and animal models was reported in three recent studies[18–20]. While all of these studies corroborate each other, our work complements the other studies in several important ways. First, the use of external spike-in control allowed us to detect global transcriptome down-regulation caused by the collateral activity of RfxCas13d. Second, we showed that collateral activity resulted in the collapse of chromatin, which may contribute to the global down-regulation of gene expression and the inhibition of cell growth. Lastly, while all other studies focused on minimizing the collateral activity, we harnessed it for cell targeting and demonstrated in a co-culture assay. Importantly, systematic comparisons between several Cas13 systems, including RfxCas13d, PspCas13b, LwaCas13a, as well as several engineered Cas13 variants revealed that RfxCas13d we used has the strongest collateral activity in mammalian cells[18–20]. While these results support the use of RfxCas13d for RNA sequence guided cell targeting, further optimization, including gRNA efficiency, specificity, and in vivo delivery systems are needed to enhance the collateral activity for selective killing of cells of interest in vivo, an important step towards marker RNA-guided elimination of cancer cells and other types of pathogenic cells.

## Methods

**Cell culture**. HEK293T, MEF-1, and U87 cell lines were used in this research. HEK293T was purchased from ATCC. MEF-1, and U87 cell lines were gifts from Muredach P. Reilly, Peter A. Sims, respectively. HEK293T, MEF-1, and U87 cells were cultured in DMEM with 4.5 g/L D-Glucose, supplemented with 10% fetal bovine serum (FBS, heat inactivated, Gibco, 10438026), no antibiotic was added. Cells were passaged upon reaching 80–90% confluency. All cells were cultured at 5% $CO_2$ and 37 °C. All cells were routinely tested for mycoplasma contamination using MycoAlert™ Mycoplasma Detection Kit (Lonza, LT07-418).

**Plasmids**. RfxCas13d-2A-GFP was expressed using the plasmid pXR001: EF1a-CasRx-2A-EGFP (Addgene #109049, ref. [6]). In the competitive growth assay described in Fig. 3f, GFP was replaced with puroR. Guide RNAs (gRNAs) were cloned into the plasmid RfxCas13d gRNA cloning backbone (Addgene #109053, ref. [6]) using oligos listed in Supplementary Table 1. DsRed plasmid was modified from pLenti-DsRed_IRES_EGFP (Addgene #92194, ref. [21]) by deleting IRES-EGFP. BFP was expressed with a plasmid originally used for the CRISPRi gRNA expression (Addgene # 62217, ref. [22]).

**gRNA design**. Ten DsRed gRNAs were designed manually. These gRNAs target sites are roughly evenly distributed in the 5′ UTR region of the DsRed reporter

while avoiding highly structured regions, a guideline commonly used for other Cas13 systems. Guide RNAs for endogenous genes were designed using an online tool (https://cas13design.nygenome.org/) as described in ref. [23].

**Plasmid transfection**. All plasmids were prepared using the NuceloSpin Plasmid Transfection-grade kit (MACHEREY-NAGEL, 740490.250). Plasmids were transfected with Lipofectamine 3000 (Invitrogen, L3000015) according to the manufacturer's protocol. Briefly, HEK293T cells were seeded at a density of $6 \times 10^5$ per well in a 6-well plate and transfected the next day. Opti-MEM I Reduced-Serum Medium (Gibco, 31985062) was prewarmed at 37 °C water bath for 30 min prior to transfection. For each well, 7.5 μl lipofectamine 3000 reagent was mixed with 125 μl Opti-MEM I Reduced-Serum Medium, then mixed with 2500 ng DNA in 125 μl Opti-MEM I Reduced-Serum Medium. After 15 min at room temperature, the mixture was added to cells. Media was not changed after transfection. Cells were harvested 48 h after transfection for gene expression analysis, western blot, flow cytometry, and RT-qPCR assays.

**siRNA transfection**. A non-targeting control siRNA pool (Horizon Discovery, D-001810-10-05), human *FTH1* siRNA pool (Horizon Discovery, L-019634-00-0005), human *HNRNPA2B1* siRNA pool (Horizon Discovery, L-011690-01-0005), human *ACTG1* siRNA pool (Horizon Discovery, L-005265-00-0005) and human *MIF* siRNA pool (Horizon Discovery, L-011335-00-0005), were transfected using Lipofectamine™ RNAiMAX Transfection Reagent (Invitrogen, 13778) according to the manufacturer's protocol. HEK293T cells were seeded at a density of $2 \times 10^5$ per well in a 6-well plate. Opti-MEM I Reduced-Serum Medium (Gibco, 31985062) was prewarmed at 37 °C water bath for 30 min prior to transfection. For each well, 7.5 μl lipofectamine RNAiMAX reagent was mixed with 125 μl Opti-MEM I Reduced-Serum Medium and then mixed with 25 pmol siRNA in 125 μl Opti-MEM I Reduced-Serum Medium. After 5 min at room temperature, the mixture was added to cells. Media was not changed after transfection. HEK293T cells were collected 48 h after transfection for RNA extraction and RT-qPCR.

**Quantitative reverse transcription PCR (RT-qPCR)**. For spike-in RT-qPCR, viable treated cells were counted and mixed with 5% (by number) untransfected mouse MEF-1 cells. Total RNA was isolated using the RNA isolation kit (MACHEREY-NAGEL, 740984.250). Synthesis of cDNA was done using the SuperScript™ IV Reverse Transcriptase (Invitrogen, 18090050). Quantitative RT-qPCR was performed with Real-Time PCR System (Applied Biosystems, Quant-studio 7 Flex), using SYBR Green Master Mix (Applied Biosystems, A25741). Relative transcript abundance was normalized to mouse *Gapdh* (*mGapdh*), *MT-CO2*, or *ACTB*. Primer sequences can be found in Supplementary Table 2.

**Poly(A) RNA-seq**. HEK293T cells were seeded at a density of $6 \times 10^5$ per well in a 6-well plate. $2 \times 10^5$ MEF-1 cells were seeded in a 6-well plate. For the experiment described in Fig. 1g, 625 ng of each plasmid (RfxCas13d-2A-GFP, DsRed, BFP, and control or DsRed-targeting gRNA) were transfected into HEK293T cells according to the above plasmid transient transfection protocol (2500 ng DNA total per well). For Fig. 1h, 5000-fold less DsRed plasmid was used (0.125 ng). Cells were harvested 48 h after transfection. For each sample, $5 \times 10^4$ MEF-1 cells were spiked into $1 \times 10^6$ viable HEK293T cells, followed by RNA extraction, poly(A) RNA pulldown, and sequencing library construction using Illumina TruSeq chemistry. Libraries were then sequenced using Illumina NovaSeq 6000 at Columbia Genome Center. RTA (Illumina) was used for base calling and bcl2fastq2 (version 2.19) for converting BCL to fastq format, coupled with adaptor trimming. Gene expression was quantified using kallisto (0.44.0) and an index containing the human transcriptome, the mouse transcriptome, and the transcripts of BFP, DsRed, and RfxCas13d-2A-GFP, as well as gRNAs. Three replicates were used for the data presented in Fig. 1g and two replicates for other RNA-seq data. Gene expression values were highly correlated between replicates (Pearson correlation coefficient $r > 0.99$) and the average values were used for downstream analysis. The RNA-seq data for siRNA-based knockdown of *HNRNPA2B1* and *HNRNPL* were obtained from previous publications[24,25].

**Western blot**. HEK293T cells were seeded at a density of $6 \times 10^5$ per well in a 6-well plate. A total of 2500 ng DNA (625 ng RfxCas13d-2A-GFP, 625 ng DsRed, 625 ng BFP, and 625 ng control or DsRed-targeting gRNA plasmids) were transfected into HEK293T cells according to the above plasmid transient transfection protocol. Cells were harvested 48 h after transfection. Cells were gently washed by cold PBS and lysed with RIPA buffer (Sigma, R0278) supplemented with protease

inhibitors (Roche, 4693132001). Cell lysates were cleared by centrifugation for 15 min at $12,000 \times g$, 4 °C. Protein lysates were loaded with LDS Sample Buffer (Invitrogen, NP0007) and Reducing Agent (Invitrogen, NP0004) after heating at 70 °C for 10 min. Proteins were separated by SDS -PAGE and subsequently transferred to PVDF membrane. Membranes were blocked in PBS with 5% non-fat milk and 0.1% Tween-20 for 1 h at room temperature and probed with appropriate primary antibodies overnight at 4 °C. Primary antibodies used: HA (Sigma, H9658, 1:1000), β-Actin (Sigma, A5441, 1:2000), LC3 (Proteintech, 14600-1-AP, 1:1000). HRP conjugated secondary antibodies were incubated for 1 h at room temperature. Blots were imaged using West Pico PLUS Chemiluminescent Substrate (Thermo fisher, 34580).

**Cell proliferation assays (EdU assays)**. HEK293T cells were seeded at a density of $3 \times 10^5$ per well in a 12-well plate with Poly-D-Lysine coated German glass coverslips (Neuvitro, H-15-PDL). A total of 1000 ng plasmids were transfected for 48 h, and then cells were incubated with EdU for 4 h, cells were subsequently fixed by 4% paraformaldehyde (PFA). EdU assays were completed with Click-iT Plus EdU Cell Proliferation Kit (Invitrogen, C10640) according to the manufacturer's protocol.

**Cell proliferation assays (WST-1)**. For plasmid transfection, HEK293T cells were seeded at a density of $1.8 \times 10^4$ per well in a 48-well plate. For siRNA transfection, HEK293T cells were seeded at a density of $1.0 \times 10^4$ per well in a 48-well plate. Cells were transfected with 250 ng DNA or 2.5 pmol siRNA per well for the indicated time of figures. Cells were incubated with the WST-1 reagent (Sigma, 11644807001) for 2 h. The absorbance of the samples was measured using a microplate reader at 440 nm.

**Apoptosis assays**. Annexin V and DAPI staining assays were performed using the apoptosis detection Kit (BioLegend, 640930) according to the manufacturer's protocol. Briefly, 48 h after transfection, cells were washed twice with cold cell staining buffer (Biolegend, 420201) and resuspended in binding buffer. APC-Annexin V and DAPI (Alternative for 7-AAD of the kit) were used for staining. Cells were incubated for 15 min at room temperature and analyzed by flow cytometry. Data were analyzed using FCS Express 7.10.

For cleaved Caspase 3 staining assays, HEK293T cells were seeded at a density of $3 \times 10^5$ per well in a 12-well plate with Poly-D-Lysine coated German glass coverslips. Cells were fixed by 4% PFA 48 h after transfection. Cleaved Caspase 3 antibody (Cell signaling technology, 9661T, 1:500) was used for detecting the expression of cleaved Caspase 3. Confocal images were captured with an LSM T-PMT confocal laser-scanning microscope (Carl Zeiss).

**Lentivirus and stable cell line generation**. We generated lentivirus with pCMV-dR8.91 and pMD2.G packaging system (gifts from Jonathan Weissman). Briefly, packaging vectors were transfected into HEK293T cells using Lipofectamine 3000 according to the manufacturer's protocol. Lentiviruses were harvested 48 h post transfection, aliquoted and stored at −80 °C for future use. For U87 stable cell lines generation, cells were seeded in a 6-well plate and infected by lentivirus with 10 μg/ml of polybrene for 48 h. Stable cells were then isolated by antibiotic selection or FACS. RfxCas13d RNA expression levels were quantified by RT-qPCR to ensure that samples had comparable expression levels. All experiments involving stable cell lines were performed with low passage cells.

**In vitro transcription**. In vitro transcription RNA was synthesized using T7 RNA synthesis kit (New England Biolabs, E2050S). Briefly, T7 primer (GAAATTAATA CGACTCACTATAGGG), DsRed gRNA with RfxCas13d pre-gRNA scaffold (CA TCCACGCTGTTTTGACCTCCGTTTCAAACCCCGACCAGTTGGTAGGGGTT CGGTGTTTCCCCTATAGTGAGTCGTATTAATTTC), and control gRNA with RfxCas13d pre-gRNA scaffold (GGAGGTCAAAACAGCGTGGATGGTTTCAAA CCCCGACCAGTTGGTAGGGGTTCGGTGTTTCCCCTATAGTGAGTCGTATT AATTTC) were ordered from IDT, 1 μg template DNA was used for 20 μl reaction, incubate at 37 °C for 2 h. The reaction was diluted with 30 μl nuclease-free water, 2 μl DNase I (RNase-free) was added, incubate at 37 °C for 15 min to remove the template DNA. Followed by RNA purification with RNA cleanup kit (New England Biolabs, T2050L). RNAs were aliquoted and stored at –80 °C for future use.

**Competitive growth assay**. A human U87 glioblastoma cell line stably expressing RfxCas13d (no GFP fusion) was generated using lentiviral integration followed by puromycin selection. Two cell lines were subsequently derived using lentiviral integration of either GFP or BFP followed by FACS. In the absence of treatment, the two cell lines, termed U87$^{GFP}$ cells and U87$^{BFP}$ cells respectively, have similar doubling time. The U87$^{GFP}$ cell line additionally expresses the DsRed reporter. The two cell lines were mixed at 1:1 ratio on day 0, then an in vitro transcribed gRNA targeting DsRed was transfected using Lipofectamine 3000 according to the manufacturer's protocol. Transfection of a non-targeting gRNA was used as a control. The fraction of U87$^{GFP}$ and U87$^{BFP}$ cells was determined using flow cytometry by detecting the GFP and BFP expression.

**Nuclear morphology quantification**. HEK293T cells were seeded $3 \times 10^5$ per well in a 12-well plate with Poly-D-Lysine coated German glass coverslips (Neuvitro, H-15-PDL). 333.3 ng RfxCas13d-2A-GFP, 333.3 ng DsRed, and 333.3 ng control or DsRed-targeting gRNA plasmids were transfected for 48 h, cells were subsequently fixed by 4% paraformaldehyde (PFA) for 15 min at room temperature. Permeabilized with 0.3% TritonX-100 for 15 min at room temperature. DAPI 1 μg/ml incubated cells for 15 min at room temperature in dark. Confocal 2D images were captured with an LSM T-PMT confocal laser-scanning microscope (Carl Zeiss). The convexity and circularity of nuclei were calculated using ImageJ with the NucleusJ plugin[26]. We measured 208 cells from the control group and 224 cells from DsRed-targeting gRNA group.

**Statistics and Reproducibility**. In general, at least three biological replicates were performed for each experiment. All statistical analyses (except the RNA-seq data analysis) were performed using the GraphPad Prism software 9.0. Two-tailed Student's $t$-test was used for comparison between two groups. ANOVA test was performed for comparing two or more than two conditions. Results in graphs are expressed as mean ± standard deviation. Kolmogorov–Smirnov test was used for the RNA-seq data analysis presented in Fig. 2d and Supplementary Fig. 5.

**Reporting summary**. Further information on research design is available in the Nature Portfolio Reporting Summary linked to this article.

## Data availability

RNA-seq data is deposited in Gene Expression Omnibus (GEO) with the accession number GSE155134. Raw gel images for Fig. 1 and Supplementary Fig. 7c are included in Supplementary Fig. 9. The numerical source data for main figures is included in Supplementary Data 1. Guide RNA sequences are included in Supplementary Table 1. Primer oligo sequences are included in Supplementary Table 2. All other data are available from the corresponding author on reasonable request.

## Code availability

Scripts for RNA-seq data analysis are available at: https://github.com/xuebingwu/cas13-collateral-activity-code (https://doi.org/10.5281/zenodo.7726352)[27].

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

## Acknowledgements
We thank Muredach Reilly and Peter Sims for generously sharing cell lines. M.R.M. is supported by an American Heart Association postdoctoral fellowship. X.W. is supported by NIH grant 1DP2GM140977 and Pershing Square Sohn Cancer Research Alliance. This research was funded in part through the NIH/NCI Cancer Center Support Grant P30CA013696 and used the Genomics and High Throughput Screening Shared Resource and CCTI Flow Cytometry Core. The CCTI Flow Cytometry Core is supported in part by the Office of the Director, National Institutes of Health under awards S10RR027050 and S10OD020056. The content is solely the responsibility of the authors and does not necessarily represent the official views of the National Institutes of Health.

## Author contributions
P.S., M.R.M., and X.W. conceived the project. P.S. performed all experiments and data analysis with assistance from M.R.M., A.O.A., Z.F., Z.C., J.S.K., A.T., and Y.G., P.S. and X.W. drafted the manuscript with input from all authors.

## Competing interests
The authors declare no competing interests.
