## [Peer Review File · Communications Biology]

Reviewers' comments:

Reviewer #1 (Remarks to the Author):

In the manuscript "RNA-guided cell targeting with CRISPR/RfxCas13d collateral activity in human cells", the authors demonstrated collateral activity of the CRISPR-RfxCas13d system in human cells, especially when reporters (i.e. DsRed) or highly expressed endogenous mRNAs were targeted. Although the use of this collateral activity to suppress the transition of cell states is an interesting and fresh idea, this reviewer has some concerns that need to be clarified or addressed regarding this concept.

Major comments:

1) In Figure 1 and Extended data figures 1-4, the authors demonstrated the collateral activity by CRISPR-RfxCas13d only after on-target activation, especially targeting highly expressed mRNAs, either reporters (i.e. DsRed) or endogenous mRNAs (FTH1, HNRNPA2B1, ACTG1 and MIF). Although they have included the analysis of rRNA degradation by Bioanalyzer, RNA integrity numbers, GFP fluorescence decay and mRNA level quantification with three different gRNAs for some targets, this reviewer would suggest to analyze the knockdown efficiency (mRNA levels quantification by qPCR) and collateral activity (GFP decay, rRNA 28S degradation, and RIN) with 2-3 different gRNAs for each endogenous target analyzed in this study and include them in extended data to ultimately demonstrate that collateral activity does not depend on specific gRNAs (structure, binding site along the transcript, etc.).

2) Although some reports came out recently showing collateral activity for CRISPR-RfxCas13d when used in vivo (PMIDs: 35953673 and bioRxiv 2022.01.17.476700) or ex vivo (PMID: 35244715), there are also others that demonstrate a specific on-targeting without any off-targets or collateral activity in different animal models such as zebrafish (PMID: 32768421), fruit fly (PMID: 33203452) or mouse (PMID: 35044815). The authors nicely used spike-in controls in RNAseq and qPCR experiments while other reports did not. Anyway, it seems that trans-activation of RfxCas13d and collateral degradation are dependent on cell types, Cas13d expression levels and delivery strategies employed. One can expect that a completely different kinetic using RNP or transfected Cas13d mRNA (consumption of molecules over time) from that observed when Cas13d is expressed by a constitutive promoter (accumulation of molecules during time) would have a different impact on collateral activity.

According to this issue and to further support the statements of this manuscript, this reviewer considers necessary to analyze the collateral activity (i.e., using DsRed/GFP reporters) using RNP delivered via adenovirus or Cas13d mRNA and gRNAs transfection.

By the way, this experiment will help to clarify that CRISPR-RfxCas13d system can still be used in vivo or ex vivo with the appropriate Cas13d kinetics and controls, such as spike-in with mouse RNA.

3) In the manuscript, the authors mentioned "a positive correlation between on-target knockdown efficiency and collateral degradation of GFP mRNA" (Extended data figure 4). To reinforce the hypothesis, this reviewer encourages to perform a R-Pearson correlation assay and represent the obtained results in an extra panel of the figure.

4) The authors said in the manuscript several times that collateral activity of Cas13d resulted in the collapse of the chromatin although they only show one microscope image with a relative collapse of the chromatin in a DAPI staining. Thus, it is needed to systematically analyze this process (i.e. quantification of this phenomenon with a higher number of cells and replicates).

5) One of the main concerns of this reviewer is the use of CRISPR-RfxCas13d as a tool to inhibit or reduce proliferation of specific cell types and suppress the transition of cell states. It is clearly

demonstrated that depletion of DsRed in U87 glioblastoma cells, lead to a high reduction of the cell population. However, an ex vivo experiment with only one cancer cell type is not sufficient to state that Cas13d can be used to cancer cells depletion. To prove this hypothesis, it would be necessary to perform in vivo experiments with cancer models in mice. Since this would require too much time and effort for this paper, at least it is mandatory a transcriptomic analysis to clearly demonstrate that deregulated genes due to collateral activity are only related to proliferation.

This reviewer understands and appreciates the effort of performing an RNAseq with PBMC system, but maybe after this time of review the authors now have more data analyzed which would aim to reinforce their hypotheses. Otherwise, this reviewer would suggest mentioning the future possibility of using the CRISPR-RfxCas13d system to deplete cell proliferation after additional experiments and controls rather than stating that can be used right now for this purpose.

6) Similarly, the authors demonstrated that one single gRNA targeting IL-6 lead to downregulation of other ten cytokines. But targeting other cytokines apart from IL-6 does also lead to a diminished inflammatory response? And even more important, again a bulk RNAseq to analyze the whole transcriptome is necessary to convince this reviewer that CRISPR-RfxCas13d can be used as a tool to suppress pro-inflammatory response or cell transitions.

Minor comments:

7) In Methods section, it is mentioned that a NES version of RfxCas13d was constructed. Nevertheless, this version has not been used in the manuscript. Please remove this information. Or even better, it would be also interesting to see whether a NES-Cas13d shows same or different on-target efficiency and collateral activity than Cas13d-NLS on high expressed mRNAs.

8) Regarding the statistics, there are some issues that still need to be reviewed. For instance, one-way ANOVA was used for Extended data figure 2 d and e, and two-way ANOVA was used for f panel. This reviewer also recommends analysing the two-side variance of all the experiments. During the whole manuscript, the authors have employed ANOVA test when more than two conditions are analysed or T-Student test when only two conditions are compared (i.e. Control vs knockdown). However, in Extended data figure 4 legend is indicated that one-way ANOVA was used for panel a, when only two conditions are compared. And finally, in Extended data figure 6, it is not indicated the meaning of asterisks. It is supposed to be * for $p < 0.05$ and ** for $p < 0.01$ according to the rest of the figures, but please indicate that in the legend.

9) In some figure legends such as Extended data figure 8 and 9, subheading are not in bold letters. Please, review the manuscript accordingly to maintain a cohesion along the text.

10) In Extended data figure 9, two different cell types are used and indicated in the figure legend. Please indicated the type of cell used also in each panel, it would be easier to perfectly understand this figure.

11) Cell death, apoptosis and autophagy are analysed to demonstrate that collateral activity induced when DsRed reporter mRNA is depleted only affects cell proliferation and DNA replication. However, when IL6 is targeted, neither apoptosis nor autophagy were analysed. It would be a nice control to reinforce the statement that CRISPR-RfxCas13d could be used to inhibit proliferation of specific cells without affecting viability.

Reviewer #2 (Remarks to the Author):

CRISPR-Cas13d originally in bacterial system show collateral activity, while initial studies in human cells and vertebrates precautionarily show that this system is not having collateral activities for different tested genes. Later reports contradict the "No collateral activity" claims, where they have shown strong collateral activity using different reporter assays. I appreciate the authors' efforts to explore the collateral activity of the cas13d system in mammalian cells using different reporter assays and 11 different endogenous genes of varying expression levels. Later authors have tried to harness this collateral activity of the Cas13d system to inhibit cell proliferation or suppress cell state transition.

This study can make a significant impact and helps the scientist to use the Cas13d system in a more effective manner, but authors need to be cautious about saying "the CRISPR/RfxCas13d system has strong collateral activity in human cells". As authors only focused on the collateral activity of Cas 13d, they need to mention it clearly that they are getting only collateral activity, or sometimes the specific activity, when targeting genes with different expression levels. If authors want to use this "the CRISPR/RfxCas13d system has strong collateral activity in human cells" they need to show that whatever gRNA they are designing gives strong collateral activity in mammalian cells, if not they need to rephrase their sentences about Cas13d collateral activity accordingly.

The main concern for reviewers about this study are:

Since this study is more focused on contradicting the earlier findings of specific activity of Cas13d system and harnessing cas13d collateral activity. Therefore, to make this manuscript more effective and put a strong point forward, authors need to be very careful with their statements and they need to mention:

1. The rules of gRNA design for cas13d are not well established, therefore authors need to mention, how they have designed the gRNAs and how many gRNAs they have tested for each gene. No information is available in the manuscript about this.

2. At least for the very first results: The DsRed reporter genes: Authors need to show how many gRNA they have tested for DsRed reporter assay, was there any correlation among different DsRed gRNA, knockdown efficiency, and collateral activity in human cells.

3. 11 endogenous genes: To prove that the high expression level of endogenous genes is showing strong collateral activity consistently with all the gRNA tested for each gene. Authors need to show the RNA integrity (Bioanalyzer data) gel (as shown in extended figure 4b) with extra ribosomal bands for other highly expressed targeted endogenous genes.

4. While harnessing cas13d collateral activity "to suppress a concerted transcriptional activation program which in turn block cell state transition" the authors need to show the effect of targeting IL-6 on the cell proliferation, DNA replication activity, and nuclei shape.

5. Reviewer's critical concern is with this experiment: Authors, in different RNA-seq experiments, suggest global repression and decrease of the whole transcriptome but the authors while harnessing the collateral activity of cas13d in the case of IL-6 knockdown, did not talk about the global effect on whole transcriptome levels, genes related to other immune response (other than cytokines), apoptosis, necrosis, and other cell death program and finally genes related with cell cycle progression and transition. The authors need to address this part in detail.

Reviewers' comments:

Reviewer #1 (Remarks to the Author):

In the manuscript "RNA-guided cell targeting with CRISPR/RfxCas13d collateral activity in human cells", the authors demonstrated collateral activity of the CRISPR-RfxCas13d system in human cells, especially when reporters (i.e. DsRed) or highly expressed endogenous mRNAs were targeted. Although the use of this collateral activity to suppress the transition of cell states is an interesting and fresh idea, this reviewer has some concerns that need to be clarified or addressed regarding this concept.

Major comments:

1) In Figure 1 and Extended data figures 1-4, the authors demonstrated the collateral activity by CRISPR-RfxCas13d only after on-target activation, especially targeting highly expressed mRNAs, either reporters (i.e. DsRed) or endogenous mRNAs (FTH1, HNRNPA2B1, ACTG1 and MIF). Although they have included the analysis of rRNA degradation by Bioanalyzer, RNA integrity numbers, GFP fluorescence decay and mRNA level quantification with three different gRNAs for some targets, this reviewer would suggest to analyze the knockdown efficiency (mRNA levels quantification by qPCR) and collateral activity (GFP decay, rRNA 28S degradation, and RIN) with 2-3 different gRNAs for each endogenous target analyzed in this study and include them in extended data to ultimately demonstrate that collateral activity does not depend on specific gRNAs (structure, binding site along the transcript, etc.).

Response: We apologize for not being clear on whether collateral activity should depend on specific gRNAs. We have now included clarifications and new data in the revised manuscript (line 45-48). In principle, all gRNAs that exhibit on-target activity should also exhibit collateral activity, because the same active site on Cas13 is used to cleave both target RNAs and by-stander RNAs (ref 2). Therefore, one would predict that if the on-target activity is highly variable across different gRNAs, then collateral activity will also be highly variable. It has been well-established for all CRISPR/Cas systems tested, including various Cas13 systems, that on-target efficacy is highly variable across gRNAs and depends on target RNA structure and whether the binding site is in coding sequence, UTR, or intron (e.g., ref 21). Indeed, we now show that with 10 gRNAs targeting DsRed, while all gRNAs led to detectable DsRed knockdown and GFP collateral degradation, both the on-target DsRed knockdown efficiency and the collateral GFP degradation are highly variable, and are highly correlated ($r=0.89$, Fig. 1e). We observed similar results for gRNAs targeting two endogenous transcripts ACTG1 and HNRNPA2B1 (Extended Data Fig. 4l-m). We therefore concluded that similar to on-target knockdown efficiency, to what extent the collateral activity of Cas13 can be detected also depends on specific gRNAs. For studies harnessing strong collateral activity for manipulating cell proliferation, it will be necessary to optimize gRNA efficiency, guided by computational tools developed for gRNA design, such as <https://www.rnatargeting.org/>.

2) Although some reports came out recently showing collateral activity for CRISPR-RfxCas13d

when used in vivo (PMIDs: 35953673 and bioRxiv 2022.01.17.476700) or ex vivo (PMID: 35244715), there are also others that demonstrate a specific on-targeting without any off-targets or collateral activity in different animal models such as zebrafish (PMID: 32768421), fruit fly (PMID: 33203452) or mouse (PMID: 35044815). The authors nicely used spike-in controls in RNAseq and qPCR experiments while other reports did not. Anyway, it seems that trans-activation of RfxCas13d and collateral degradation are dependent on cell types, Cas13d expression levels and delivery strategies employed. One can expect that a completely different kinetic using RNP or transfected Cas13d mRNA (consumption of molecules over time) from that observed when Cas13d is expressed by a constitutive promoter (accumulation of molecules during time) would have a different impact on collateral activity.

According to this issue and to further support the statements of this manuscript, this reviewer considers necessary to analyze the collateral activity (i.e., using DsRed/GFP reporters) using RNP delivered via adenovirus or Cas13d mRNA and gRNAs transfection.

By the way, this experiment will help to clarify that CRISPR-RfxCas13d system can still be used in vivo or ex vivo with the appropriate Cas13d kinetics and controls, such as spike-in with mouse RNA.

Response: We agree with the reviewer that the collateral activity of the Cas13d system can potentially be mitigated, e.g., by using alternative delivery methods. While reducing Cas13 collateral activity is the primary focus of several other manuscripts (ref 16-18), our study focuses on harnessing the collateral activity for cell targeting. Furthermore, as explained below, we do not anticipate RNP/RNA delivery to behave much differently from DNA transfection. A key feature of the RNP-based delivery is that the delivered protein/RNA is short-lived. However, CRISPR/Cas13 is inherently auto-inhibitory and short-lived. Once expressed and activated by target transcripts, the collateral activity degrades Cas13 mRNA and gRNA themselves. This is evident in Fig. 1d: in the presence of DsRed and a targeting gRNA, Cas13 protein is almost completely gone within 24 hours of transfection. We thus do not expect switching to RNP delivery will cause a large difference. Indeed, our competitive growth assay in U87 cells (Fig. 3f) was performed using in vitro transcribed gRNAs, and we still observed strong collateral activity and cell proliferation defect. In this experiment, Cas13 protein is expressed stably. However, the transfected, in vitro transcribed gRNA is short-lived and is limiting even though Cas13 protein may be in excess.

3) In the manuscript, the authors mentioned “a positive correlation between on-target knockdown efficiency and collateral degradation of GFP mRNA” (Extended data figure 4). To reinforce the hypothesis, this reviewer encourages to perform a R-Pearson correlation assay and represent the obtained results in an extra panel of the figure.

Response: We have now added the suggested analysis for both DsRed (Fig. 1e) and two endogenous targets (Extended Data Fig. 4l-m).

4) The authors said in the manuscript several times that collateral activity of Cas13d resulted in

the collapse of the chromatin although they only show one microscope image with a relative collapse of the chromatin in a DAPI staining. Thus, it is needed to systematically analyze this process (i.e. quantification of this phenomenon with a higher number of cells and replicates).

Response: We have now included quantification of nuclear morphology in 224 targeted cells and 208 control cells by using the NucleusJ plugin in ImageJ (Fig. 3d). Specifically, for each nucleus, the software calculated its convexity and circularity, which measure the overall smoothness of the nuclear outline and how similar it is to a circle, respectively. For both measurements, there is a significant decrease in targeted cells ($P < 0.0001$).

5) One of the main concerns of this reviewer is the use of CRISPR-RfxCas13d as a tool to inhibit or reduce proliferation of specific cell types and suppress the transition of cell states. It is clearly demonstrated that depletion of DsRed in U87 glioblastoma cells, lead to a high reduction of the cell population. However, an ex vivo experiment with only one cancer cell type is not sufficient to state that Cas13d can be used to cancer cells depletion. To prove this hypothesis, it would be necessary to perform in vivo experiments with cancer models in mice. Since this would require too much time and effort for this paper, at least it is mandatory a transcriptomic analysis to clearly demonstrate that deregulated genes due to collateral activity are only related to proliferation.

This reviewer understands and appreciates the effort of performing an RNAseq with PBMC system, but maybe after this time of review the authors now have more data analyzed which would aim to reinforce their hypotheses. Otherwise, this reviewer would suggest mentioning the future possibility of using the CRISPR-RfxCas13d system to deplete cell proliferation after additional experiments and controls rather than stating that can be used right now for this purpose.

Response: We agree with the reviewer that additional experiments, including systematic characterization of transcriptome changes and optimization of delivery methods are needed to achieve the goal of selective depletion of a specific cell type, especially for in vivo applications. In the revised manuscript we have stressed the importance of further development of the proposed application in future studies (line 149-153). Regarding the proliferation defect resulted from DsRed targeting in U87 cells, we do not anticipate that only proliferation related genes are affected by the collateral activity. Our RNA-seq data in HEK293T cells, which was generated using the same gRNA and target (DsRed), show that almost all mRNAs are affected by the collateral activity (Fig. 1g), not just proliferation-related genes. While alteration of proliferation is an important phenotype that we have focused on, many other cellular processes are likely also affected.

6) Similarly, the authors demonstrated that one single gRNA targeting IL-6 lead to downregulation of other ten cytokines. But targeting other cytokines apart from IL-6 does also lead to a diminished inflammatory response? And even more important, again a bulk RNAseq to analyze the whole transcriptome is necessary to convince this reviewer that CRISPR-RfxCas13d can be used as a tool to suppress pro-inflammatory response or cell transitions.

Response: We agree with the reviewer that more systematic analyses are needed for the IL-6 targeting part of our study. However, we have not generated the mentioned RNA-seq data using human primary PBMCs due to technical challenges. Given that the IL-6 targeting part is not essential for our main claims, and the urgency to publish our results in light of the publication of similar studies during the peer review of our manuscript, we have discussed with the editor and we have decided to drop the IL-6 section. The manuscript will still include systematic analysis of collateral activity using reporters and endogenous genes, the effect on cell proliferation, and the proof-of-principle experiment depleting one U87 cell population from a mixture.

Minor comments:

7) In Methods section, it is mentioned that a NES version of RfxCas13d was constructed. Nevertheless, this version has not been used in the manuscript. Please remove this information. Or even better, it would be also interesting to see whether a NES-Cas13d shows same or different on-target efficiency and collateral activity than Cas13d-NLS on high expressed mRNAs.

Response: We apologize for the error. We have now removed the relevant text in the Methods section. Our preliminary analyses showed that NES-Cas13d resulted in slightly weaker collateral degradation of GFP when DsRed is targeted, consistent with a slightly weaker on-target knockdown efficiency. However, we have not performed systematic analysis using RNA-seq, RIN, or endogenous targets.

8) Regarding the statistics, there are some issues that still need to be reviewed. For instance, one-way ANOVA was used for Extended data figure 2 d and e, and two-way ANOVA was used for f panel. This reviewer also recommends analysing the two-side variance of all the experiments.

During the whole manuscript, the authors have employed ANOVA test when more than two conditions are analysed or T-Student test when only two conditions are compared (i.e. Control vs knockdown). However, in Extended data figure 4 legend is indicated that one-way ANOVA was used for panel a, when only two conditions are compared.

*And finally, in Extended data figure 6, it is not indicated the meaning of asterisks. It is supposed to be * for $p < 0.05$ and ** for $p < 0.01$ according to the rest of the figures, but please indicate that in the legend.*

Response: We have revised the manuscript following these very helpful suggestions.

9) In some figure legends such as Extended data figure 8 and 9, subheading are not in bold letters. Please, review the manuscript accordingly to maintain a cohesion along the text.

Response: We have revised the manuscript following these very helpful suggestions. As explained above, after consulting with the editor, we have removed the IL-6 targeting section from the manuscript, which includes the original Extended Data Fig. 9.

10) In Extended data figure 9, two different cell types are used and indicated in the figure legend. Please indicated the type of cell used also in each panel, it would be easier to perfectly understand this figure.

Response: As explained above, after consulting with the editor, we have removed the IL-6 targeting section from the manuscript, which includes the original Extended Data Fig. 9.

11) Cell death, apoptosis and autophagy are analysed to demonstrate that collateral activity induced when DsRed reporter mRNA is depleted only affects cell proliferation and DNA replication. However, when IL6 is targeted, neither apoptosis nor autophagy were analysed. It would be a nice control to reinforce the statement that CRISPR-RfxCas13d could be used to inhibit proliferation of specific cells without affecting viability.

Response: As explained above, after consulting with the editor, we have removed the IL-6 targeting section from the manuscript.

Reviewer #2 (Remarks to the Author):

CRISPR-Cas13d originally in bacterial system show collateral activity, while initial studies in human cells and vertebrates precautionarily show that this system is not having collateral activities for different tested genes. Later reports contradict the “No collateral activity” claims, where they have shown strong collateral activity using different reporter assays. I appreciate the authors' efforts to explore the collateral activity of the cas13d system in mammalian cells using different reporter assays and 11 different endogenous genes of varying expression levels. Later authors have tried to harness this collateral activity of the Cas13d system to inhibit cell proliferation or suppress cell state transition.

This study can make a significant impact and helps the scientist to use the Cas13d system in a more effective manner, but authors need to be cautious about saying “the CRISPR/RfxCas13d system has strong collateral activity in human cells”. As authors only focused on the collateral activity of Cas 13d, they need to mention it clearly that they are getting only collateral activity, or sometimes the specific activity, when targeting genes with different expression levels. If authors want to use this “the CRISPR/RfxCas13d system has strong collateral activity in human cells” they need to show that whatever gRNA they are designing gives strong collateral activity in mammalian cells, if not they need to rephrase their sentences about Cas13d collateral activity accordingly.

Response: We have revised the manuscript accordingly. See the first paragraph in Discussion.

The main concern for reviewers about this study are:

Since this study is more focused on contradicting the earlier findings of specific activity of Cas13d system and harnessing cas13d collateral activity. Therefore, to make this manuscript

more effective and put a strong point forward, authors need to be very careful with their statements and they need to mention:

1. The rules of gRNA design for cas13d are not well established, therefore authors need to mention, how they have designed the gRNAs and how many gRNAs they have tested for each gene. No information is available in the manuscript about this.

Response: We have now included these information in Methods. Briefly, when we started this project, there is no tools for designing gRNAs for the RfxCas13d system. We thus designed ten gRNAs manually for DsRed. These gRNAs target sites roughly evenly distributed in the 5' UTR region of the DsRed reporter while avoiding highly structured regions, a guideline commonly used for other Cas13 systems. Later, the Sanjana group developed a webserver for gRNA design using machine learning models trained on high-throughput data²¹ (<https://cas13design.nygenome.org/>). The webserver was used to design gRNAs targeting endogenous transcripts.

2. At least for the very first results: The DsRed reporter genes: Authors need to show how many gRNA they have tested for DsRed reporter assay, was there any correlation among different DsRed gRNA, knockdown efficiency, and collateral activity in human cells.

Response: We have now provided the requested data in Fig. 1e and updated the text. Briefly, ten gRNAs targeting DsRed were tested, and there is a strong positive correlation between knockdown efficiency and collateral activity ($r=0.89$).

3. 11 endogenous genes: To prove that the high expression level of endogenous genes is showing strong collateral activity consistently with all the gRNA tested for each gene. Authors need to show the RNA integrity (Bioanalyzer data) gel (as shown in extended figure 4b) with extra ribosomal bands for other highly expressed targeted endogenous genes.

Response: We have now added the requested data in Extended Data Fig. 4b with 3-6 biological replicates per gene all showing an putative extra ribosomal band indicating rRNA fragmentation.

4. While harnessing cas13d collateral activity “to suppress a concerted transcriptional activation program which in turn block cell state transition” the authors need to show the effect of targeting IL-6 on the cell proliferation, DNA replication activity, and nuclei shape.

Response: We agree with the reviewer that more systematic analyses are needed for the IL-6 targeting part of our study. Given that the IL-6 targeting part is not essential for our main claims, and the urgency to publish our results in light of the publication of similar studies during the peer review of our manuscript, we have discussed with the editor and we have decided to drop the IL-6 section. The manuscript will still include systematic analysis of collateral activity using reporters and endogenous genes, the effect on cell proliferation, and the proof-of-principle experiment depleting one U87 cell population from a mixture.

5. Reviewer's critical concern is with this experiment: Authors, in different RNA-seq experiments, suggest global repression and decrease of the whole transcriptome but the authors while harnessing the collateral activity of cas13d in the case of IL-6 knockdown, did not talk about the global effect on whole transcriptome levels, genes related to other immune response (other than cytokines), apoptosis, necrosis, and other cell death program and finally genes related with cell cycle progression and transition. The authors need to address this part in detail.

Response: As explained above, after consulting with the editor, we have removed the IL-6 targeting section from the manuscript.

Reviewer #2 (Remarks to the Author):

Title: RNA-guided cell targeting with CRISPR/RfxCas13d collateral activity in human cells

Revision 2: Reviewer comments:

1. Since an important experiment supporting the title of the manuscript has been removed and only one invitro reporter assay was done to support the title of the manuscript, therefore either authors can think of slightly modifying the title or they can try targeting one of the endogenous non-essential RNA markers in a cancer cell line and recapitulate "RNA-guided cell targeting with CRISPR/RfxCas13d collateral activity in human cells".

2. The authors have satisfactorily answered the reviewers' major concerns and since they have removed the "IL-6" part from the revised manuscript, this reviewer does not have major concerns with this revised manuscript except for the above-mentioned point.

REVIEWERS' COMMENTS:

Reviewer #1 (Remarks to the Author):

In the manuscript “RNA-guided cell targeting with CRISPR/RfxCas13d collateral activity in human cells”, the authors demonstrated collateral activity of the CRISPR-RfxCas13d system in human cells, especially when reporters (i.e. DsRed) or highly expressed endogenous mRNAs were targeted. Nicely, RNAseq with spike-in mouse RNA allowed them to demonstrate a global downregulation. In the last part of the paper, the authors propose the use of this collateral activity to suppress specific cell transitions or states such as proliferative cells in cancer, which is an interesting idea.

After this review, the authors have covered almost all concerns from this reviewer, adding new data to highly support that CRISPR-RfxCas13d presents collateral activity when abundant RNAs are targeted with highly efficient gRNAs.

However, this reviewer would have really like to see more experiments about IL-6 part rather than been removed. Anyway, this reviewer understands why this particular part of the paper has been removed and recognises the effort of the authors to demonstrate the collateral activity of RfxCas13d in human cells when abundant RNAs are targeted with highly efficient gRNAs and a high expression of both Cas13d and guides. Actually, this is a crucial appreciation that need to be clearly stated within the manuscript: collateral activity of CRISPR-RfxCas13d uniquely appeared with high expression of CRISPR-Cas components (Cas protein and gRNAs), high abundant/expressed targets and highly efficient guides.

Response: we have included a similar language in the main text (line 157-160).

Nevertheless, as the interleukin targeting part was removed from the manuscript, this reviewer also consider that the title should be change prior to publication. A single experiment without a wide (i.e. transcriptomic) analysis of the cell state and response to CRISPR-RfxCas13d targeting is not enough to maintain that collateral activity can be used to deplete specific cell types or states.

Response: we have changed the title to: “The collateral activity of the CRISPR/RfxCas13d system in human cells.”

Reviewer #2 (Remarks to the Author):

Title: RNA-guided cell targeting with CRISPR/RfxCas13d collateral activity in human cells

Revision 2: Reviewer comments:

1. Since an important experiment supporting the title of the manuscript has been removed and only one invitro reporter assay was done to support the title of the manuscript, therefore either

authors can think of slightly modifying the title or they can try targeting one of the endogenous non-essential RNA markers in a cancer cell line and recapitulate "RNA-guided cell targeting with CRISPR/RfxCas13d collateral activity in human cells".

Response: we have changed the title to: "The collateral activity of the CRISPR/RfxCas13d system in human cells."

2. The authors have satisfactorily answered the reviewers' major concerns and since they have removed the "IL-6" part from the revised manuscript, this reviewer does not have major concerns with this revised manuscript except for the above-mentioned point.